# A Simulation Optimization Factor of Si(111)-Based AlGaN/GaN Epitaxy for High Frequency and Low-Voltage-Control High Electron Mobility Transistor Application

**DOI:** 10.3390/mi14010168

**Published:** 2023-01-09

**Authors:** He Guan, Guiyu Shen, Shibin Liu, Chengyu Jiang, Jingbo Wu

**Affiliations:** 1School of Microelectronics, Northwestern Polytechnical University, Xi’an 710129, China; 2School of Electronics and Information, Northwestern Polytechnical University, Xi’an 710129, China

**Keywords:** AlGaN/GaN epitaxy, HEMT, simulation

## Abstract

The effects of barrier layer thickness, Al component of barrier layer, and passivation layer thickness of high-resistance Si (111)-based AlGaN/GaN heterojunction epitaxy on the knee-point voltage (*V_knee_*), saturation current density (*I_d-sat_*), and cut-off frequency (*f_t_*) of its high electron mobility transistor (HEMT) are simulated and analyzed. A novel optimization factor *OPTIM* is proposed by considering the various performance parameters of the device to reduce the *V_knee_* and improve the *I_d-sat_* on the premise of ensuring the *f_t_*. Based on this factor, the optimized AlGaN/GaN epitaxial structure was designed with a barrier layer thickness of 20 nm, an Al component in the barrier layer of 25%, and a SiN passivation layer of 6 nm. By simulation, when the gate voltage V_g_ is 0 V, the designed device with a gate length of 0.15 μm, gate-source spacing of 0.5 μm, and gate-drain spacing of 1 μm presents a high *I_d-sat_* of 750 mA/mm and a low *V_knee_* of 2.0 V and presents *f_t_* and maximum frequency (*f_max_*) as high as 110 GHz and 220 GHz, respectively. The designed device was fabricated and tested to verify the simulation results. We demonstrated the optimization factor *OPTIM* can provide an effective design method for follow-up high-frequency and low-voltage applications of GaN devices.

## 1. Introduction

GaN as a typical third-generation semiconductor material presents good characteristics of high power density, high breakdown voltage, and high electron saturation drift speed [1,2], so it can withstand higher voltage and output higher energy density and can work at higher ambient temperatures. Due to the polarization (spontaneous polarization and piezoelectric polarization) effect, the GaN high electron mobility transistor (HEMT) can form a two-dimensional electron gas (2DEG) with a high concentration and high mobility in the potential barrier of the heterojunction interface without doping [3,4,5], which present less resistance, faster switching speed, smaller parasitic parameters, and more efficient heat dissipation compared with the traditional Si and GaAs transistors. For RF application, the commonly used substrate materials for GaN HEMTs are SiC and Si [6,7,8]. Compared with the SiC substrate, the Si substrate has a better cost advantage. Therefore, high resistance (HR) Si(111)-based AlGaN/GaN HEMTs are considered the most promising GaN RF device and have become a research hotspot of researchers in recent years [9,10,11,12,13]. 

Currently, research on Si-based GaN HEMT devices is mainly focused on high-voltage situations such as base stations or radars [14,15,16,17]. However, the power supply voltage of mobile terminal devices is less than 12 V, and the endurance capacity and cost of the devices need to be considered. Therefore, the existing GaN HEMT devices are not suitable for 5G mobile terminals, and there are relatively few papers on low-voltage applications. It is found that the epitaxial structure parameters have a great influence on the performance of devices, but the quantitative influence effect is relatively less studied. In order to further elaborate on the influence of epitaxial structure parameters on device characteristics, in this paper, we have carried out detailed simulation research on the influence of physical parameters such as the thickness of the epitaxial barrier layer, Al component of the barrier layer, and thickness of the SiN passivation layer on the epitaxial surface on device characteristics, and proposed an optimization factor *OPTIM* based on the cut-off frequency(*f_t_*), knee-point voltage, and saturated current density. Finally, a GaN HEMT device with low knee-point voltage and high *f_t_* is designed, the electrical characteristic parameters of the device are simulated and analyzed, and the device is fabricated and tested to verify the simulation results.

## 2. Simulation

In view of the lattice mismatch and thermal expansion coefficient mismatch between the Si substrate and GaN, a layer of AlN and two layers of AlGaN as the buffer layer between the Si substrate and GaN are designed. The ratio of the Al element to Ga element in the lower buffer is 0.54:0.46 and 0.3:0.7 in the upper layer. This buffer layer introduces some compressive stress to neutralize the tensile stress generated during cooling, so as to avoid the crack problem in the process of cooling to room temperature after growing the GaN channel layer. The buffer layer can also reduce the defects caused by the lattice mismatch between the Si substrate and GaN during the epitaxial growth and improve the reliability of devices. In order to enhance the polarization effect of heterojunction, an AlN insertion layer is designed between the GaN channel layer and AlGaN barrier layer to form the AlGaN/AlN/GaN heterojunction. This structure can aptly increase the depth of the heterojunction potential and enhance the polarization effect of the heterojunction. Compared with the conventional device structure, the saturation current density of the device can be improved. The specific device structure is shown in Figure 1.

Based on the above epitaxy structure, we carried out detailed simulation research on the influence of physical parameters such as the thickness of the epitaxial barrier layer, the Al component of the barrier layer, and the thickness of the SiN passivation layer on the epitaxial surface on device characteristics via the Crosslight simulation platform to build the device structure. Meanwhile, Apsys software of the platform is used to add physical models such as the polarization effect model, electron mobility model, and carrier generation composite model, and set the appropriate bias voltage. In the simulation process, the gate length of the device is taken as 0.15 μm, the gate-source spacing of the device is taken as 0.5 μm, and the gate-drain spacing is taken as 1 μm.

### 2.1. AlGaN Barrier Layer Thickness

In order to analyze the relationship between device saturation, current density, and barrier layer thickness, AlGaN/GaN HEMTs with a barrier layer thickness of 10 nm to 30 nm were simulated. The simulation curves of the channel carrier concentration and mobility as a function of the barrier thickness are shown in Figure 2. It can be seen that with the increase in the barrier layer thickness, the depth of the potential well of the heterojunction band increases, the polarization effect increases, and the carrier concentration increases. When the thickness of the barrier layer is small, as the barrier layer thickens and the carrier concentration increases, the shielding effect of coulomb force enhancement can lead to the weakening of the modulation doping scattering effect by the ionizing donor in the AlGaN barrier layer, thus increasing the mobility. However, when the barrier layer thickness exceeds 30 nm, the slope of the curve increased, meaning the mobility of carriers decreases significantly because the increase in the barrier layer thickness leads to a decrease in the electric field intensity at the channel, which becomes the dominant factor.

Simulation curves of saturation current density and transconductance of the device as a function of the barrier layer thickness are shown in Figure 3. Therefore, increasing the thickness of the barrier layer can improve the saturation current density of the device. Significantly, when the barrier thickness increases from 15 nm to 25 nm, the increase in the saturation current is larger than that when the barrier thickness increases from 40 nm to 50 nm. This is because when the barrier layer thickness exceeds 40 nm, the change in the energy band of the barrier layer and the shape of the potential well is very small, and the electron gas concentration tends to be saturated, so the increase in the saturation current becomes smaller. The transconductance of the device decreases when the barrier thickness increases. In particular, when the barrier thickness exceeds 30 nm, the increase in the barrier thickness leads to a decrease in the electric field strength at the channel, which becomes the dominant factor, and the carrier mobility and the device transconductance are significantly reduced.

The device knee-point voltage curve, along with the change in the thickness of the barrier layer, is shown in Figure 4, and the knee-point voltage visibly increases with the increasing thickness of the barrier layer. This is because as the barrier layer thickness increases, the polarization effect of the heterojunction, the 2DEG, and the saturation current density increase. The polarization effect of the heterojunction increases, the two-dimensional electron gas concentration increases, and the saturation current density increases. When the on-resistance of the device is constant, the increase in the saturation current density will lead to an increase in the knee-point voltage. Therefore, increasing the barrier layer thickness will lead to an increase in the knee-point voltage.

Simulation results of *f_t_* and maximum frequency (*f_max_*) of the device with the barrier layer thickness are shown in Figure 5. When the barrier thickness is less than 25 nm, *f_t_* and the highest oscillation frequency of the device increase with the increase in the barrier thickness. When the barrier layer thickness is 25 nm, *f_t_* and *f_max_* of the device reach 100 GHz and 180 GHz, respectively. However, when the barrier layer thickness exceeds 30 nm, *f_t_* and *f_max_* decrease significantly. When the barrier layer thickness reaches 40 nm, *f_t_* and *f_max_* decrease to 42 GHz and 85 GHz, respectively, which is caused by the obvious reduction of the transconductance of the device.

According to the above simulation analysis, increasing the thickness of the barrier layer can improve the polarization effect strength of the heterojunction, increase 2DEG, and then increase the saturation current density of the device. However, when the on-resistance of the device is constant, the increase in saturation current density will lead to an increase in the knee-point voltage, and a barrier layer thickness over 30 nm will lead to a significant reduction of the transconductance and *f_t_*. Therefore, when optimizing the barrier layer thickness and improving the saturation current density, the barrier layer thickness should be no more than 30 nm.

### 2.2. Al Component of AlGaN Barrier Layer

In order to analyze the influence mechanism of barrier layer Al components on device performance parameters, the 2DEG density and mobility of devices with different barrier layer Al components are simulated, and the simulation results are shown in Figure 6. With the increase in the Al component in the barrier layer, the polarization effect of the heterojunction is enhanced, the concentration of 2DEG in the channel increases, and the electron mobility decreases.

The simulation results of the Al component of the barrier layer on device saturation current and transconductance are shown in Figure 7. The saturation current density of the device can be increased by increasing the Al component of the barrier layer, which is caused by the increase in the 2DEG density with the increase in the Al component. Increasing the Al component of the barrier layer leads to a decrease in the transconductance of the device because increasing the Al component of the barrier layer leads to a decrease in electron mobility. The intensity of the alloy-disordered scattering caused by the random distribution of Al and Ga atoms in the barrier layer satisfies the qualitative relation of F=k·x(1−x) where, F represents the disordered scattering intensity of the alloy, and X represents the Al component in Al_x_Ga_(1-x)_N material. When x = 0.5, the alloy-disordered scattering is the strongest. With the increase in the Al component from 20% to 30%, the alloy-disordered scattering effect of the AlGaN barrier layer is gradually enhanced, leading to a gradual decline in mobility. In addition, the variation of the Al component also affects the roughness scattering at the interface of the heterojunction. With the increase in the Al component in the barrier layer, the density of 2DEG increases, and the distribution of electrons is closer to the interface of heterojunction, which is more sensitive to the roughness of the interface. At the same time, the interface roughness is also affected by the Al component. With the increase in the Al component, the interface becomes rougher. The combined effect of the above two factors significantly enhances the scattering effect of interface roughness with the increase in the Al component. When the Al component of the barrier layer exceeds 30%, the electron mobility decreases rapidly and the device transconductance decreases.

The variation curve of the knee-point voltage of the device with the Al component of the barrier layer is shown in Figure 8. It can be seen that the knee-point voltage increases with the increase in the Al component in the barrier layer. The on-resistance of the device satisfies 1/R_on_ = ΔI_d_/ΔV_d_, which is the slope of the output characteristic curve. The simulation results show that the slope of the curve increases with the increase in the Al component of the barrier layer, which means that the device conduction resistance decreases. The reason is that the 2DEG density in the channel increases with the increase in the Al component in the barrier layer. Although the increase in the Al component will enhance the interface roughness scattering and alloy-disordered scattering effect at the channel, leading to a decrease in mobility, the 2DEG density will also increase significantly. The combined effect of the mobility and 2DEG density will result in a decrease in the on-resistance of the device. The on-resistance and saturation current density determine the knee-point voltage of the device. Although the on-resistance decreases with the increase in the Al component in the barrier layer, the saturation current density increases, and the variation range is larger than that of the on-resistance. Considering the changes in the two factors, the increase in the Al components in the barrier layer will lead to an increase in the knee-point voltage.

Simulation results of *f_t_* and *f_max_* with different Al components are shown in Figure 9. The simulation results show that *f_t_* and *f_max_* decrease with the increase in the Al component of the barrier layer. When the Al component of the barrier layer exceeds 30%, the transconductance of the device decreases significantly with the increase in the Al component of the barrier layer, resulting in a significant reduction of *f_t_* and *f_max_*. 

According to the above simulation analysis, increasing the Al component of the barrier layer can improve the polarization effect strength of the heterojunction, increase the 2DEG density, and then increase the saturation current density of the device, but will lead to the increase in the knee-point voltage at the same time. In addition, when the Al component of the barrier layer exceeds 30%, the device transconductance is significantly reduced, which will lead to a significant decrease in *f_t_* and *f_max_*. Therefore, when optimizing the Al component of the barrier layer and increasing the saturation current density, it should be ensured that the Al component of the barrier layer does not exceed 30%.

### 2.3. SiN Passivation Layer Thickness Optimization

The growth of a SiN passivation layer on the surface of traditional structure devices can improve the Schottky barrier of the AlGaN/GaN heterostructure, inhibit the surface state caused by AlGaN/GaN heterostructure defects, significantly reduce the gate leakage, reduce the impact of environmental factors on the electrical performance of devices, and improve the reliability of devices. Considering the difference in the thermal expansion coefficient between SiN material and AlGaN material, cracks may occur in the cooling process after the SiN passivating layer is directly grown on the surface of the device with a traditional structure. When the gate voltage is 2 V, the gate drain current density of the device varies with the passivation layer thickness as shown in Figure 10. It can be seen that increasing the thickness of the SiN passivation layer can reduce the gate leakage current. This is because with the increase in the passivation layer thickness, the gate Schottky barrier height increases and the gate leakage of the device decreases significantly.

## 3. Optimization Factor OPTIM

According to the above analysis, increasing the barrier thickness and Al component can improve the saturation current density, but when the on-resistance is constant, the increase in the saturation current density will lead to an increase in the knee voltage at the same time, while increasing the barrier thickness and Al component will lead to the decrease in the cutoff frequency. The influence of many parameters on device performance is mutually restricted, so it is necessary to optimize the device performance through reasonable design. The devices designed in this paper are used in high-frequency and low-voltage applications, which need to improve the saturation current density and *f_t_* and reduce the knee-point voltage. In order to compromise and consider the above three parameters, the formula *OPTIM* = *I_sat_*f_t_/V_knee_* is proposed to describe the optimization degree of the barrier layer thickness and Al component. The larger the *OPTIM* value is, the better the optimization effect is. 

The *OPTIM* curves of different barrier layer thicknesses and different barrier layer Al components are shown in Figure 11. The barrier thickness of 20 nm and Al components of 25% can improve the saturation current density of the device and reduce the knee voltage as much as possible under the condition that *f_t_* and transconductance meet the requirements of the basic indicators. Because the lattice constant and thermal expansion coefficient of SiN material are different from the GaN material, the thickness of the SiN passivation layer is too high, which will introduce greater stress to the device in actual preparation, increase the difficulty of device preparation, and affect the performance and reliability of the device. In order to minimize gate leakage while ensuring device reliability, the passivation layer thickness is designed to be 6 nm.

## 4. Design and Simulation Result

According to the above design results, the barrier layer thickness of the designed HR Si-AlGaN/GaN epitaxial material is 20 nm, the Al component of the barrier layer is 25%, and the thickness of the SiN passivation layer is 6 nm. The AlGaN/GaN HEMT device with a gate length of 0.15 μm, gate-source spacing of 0.5 μm, and gate-drain spacing of 1 μm is simulated. As shown in Figure 12a, the carrier mobility of the device is 2178 cm^2^/Vs. The output characteristics are shown in Figure 12b. When the gate voltage is 0 V, the *I_sat_* is 750 mA/mm and the V_knee_ is approximately 2.0 V. The S parameter AlGaN/GaN HEMTs were simulated and based on the simulation results, the H_21_ and Unilateral Power Gain (UPG) were calculated. According to the law of −20 dB/dec, the frequency when H_21_ and UPG were equal to 1 was extrapolated to obtain the *f_t_* and *f_max_*, and as shown in Figure 12c, *f_t_* of the device was 110 GHz and *f_max_* of the device was 220 GHz. 

## 5. Epitaxy and Device Fabrication

Based on the above simulation results and design, the growth test of the AGaN/GaN epitaxy was fabricated. The test data in Table 1 show that the mobility of the fabricated AlGaN/GaN epitaxy is 2063 cm^2^⁄(V∙s), the 2DEG density is 1.86 × 10^13^ cm^−2^, and the sheet resistance is 161.41 Ω/sq. The test results are consistent with the simulation results and meet the design expectations well.

Based on the above epitaxy, the AlGaN/GaN HEMTs device was fabricated. It can be seen from Figure 13a that the gate leakage current (*I*_leakage_) of the device is as low as approximately 3 μA/mm, which means the optimized epitaxy can suppress the leakage current effectively. Figure 13b is the transmission characteristic curve of the device. The *V*_ds_ increases from 0 to 12 V with a step of 0.1 V, while the *V*_gs_ increases from −10 V to 0 with a step of 1 V. It can be seen that when the gate voltage is 0 V, the device presents an *I_sat_* of approximately 570 mA/mm, which is fundamentally similar to the simulation result of 750 mA/mm but slightly lower, and the *V*_knee_ of the device is approximately 4 V. It is found that the magnitude of the test results is essentially consistent with the simulation results, but there are minor gaps. This is primarily because the simulation results only show the ideal results, but there are many non-ideal parasitic effects in the actual devices, leading to a decline in the performance of the actual devices. In addition, the device fabrication process could affect the device performance as well, thus we will continually optimize the device fabrication process in our future studies to further improve the device performance.

## 6. Conclusions

Aiming at the application of high-frequency and low-voltage GaN HEMT devices, the effects of the barrier layer thickness, Al component of the barrier layer, passivation layer thickness, and other parameters of HR Si AlGaN/GaN HEMTs epitaxial material on the knee-point voltage, saturation current density, and *f_t_* of the device were simulated and analyzed in this paper. An optimization factor *OPTIM* based on *f_t_*, knee-point voltage, and saturated current density was proposed. The barrier layer thickness of the epitaxial material was 20 nm, the Al component of the barrier layer was 25%, and the SiN passivation layer thickness was 6 nm. The device size was 0.15 μm in gate length, 0.5 μm in gate-source spacing, and 1 μm in gate-drain spacing. Under the conditions of gate voltage V_g_ = 0 V, the saturation current density of the device was 750 mA/mm, the knee voltage was 2.0 V, *f_t_* of the device was 110 GHz, and *f_max_* was 220 GHz. The designed device was fabricated and tested to verify the simulation results. The simulation design method proposed in this paper takes into account the performance parameters of the device, reduces the knee-point voltage, and improves the saturation current density on the premise of ensuring *f_t_* of the device. It can provide an effective design reference for follow-up high-frequency and low-voltage applications of GaN devices.

## Figures and Tables

**Figure 1 micromachines-14-00168-f001:**
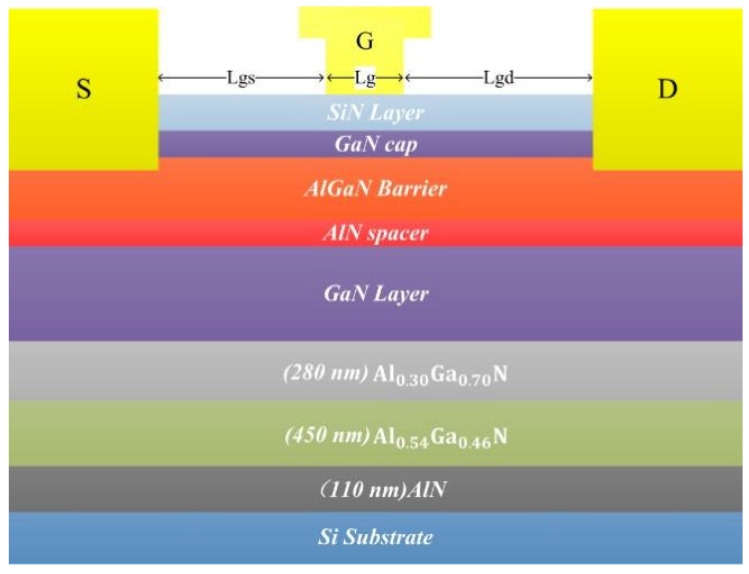
Schematic diagram of the overall structure of the device.

**Figure 2 micromachines-14-00168-f002:**
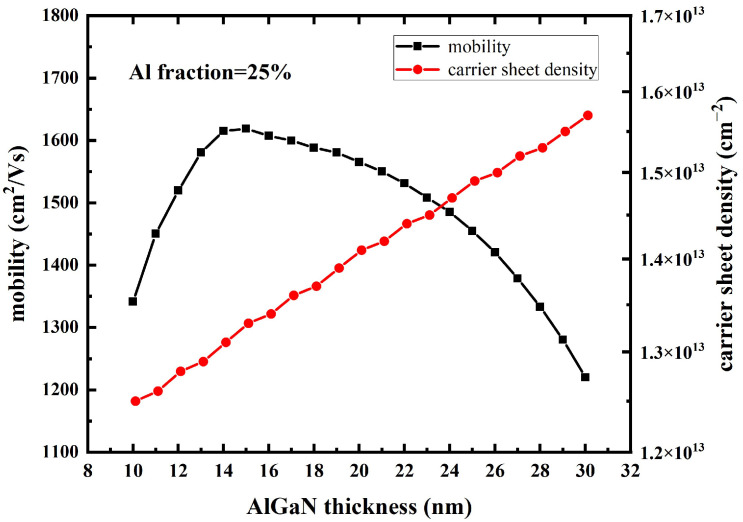
The mobility and carrier sheet density of device as AlGaN thickness variation.

**Figure 3 micromachines-14-00168-f003:**
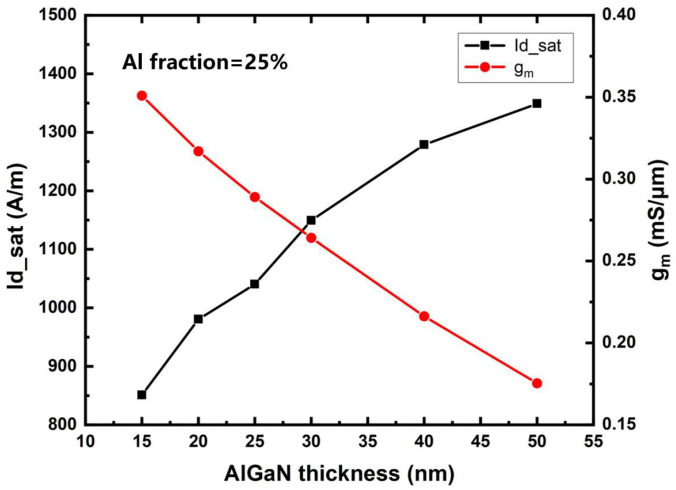
The saturation current density and transconductance of device as barrier layer thickness variation.

**Figure 4 micromachines-14-00168-f004:**
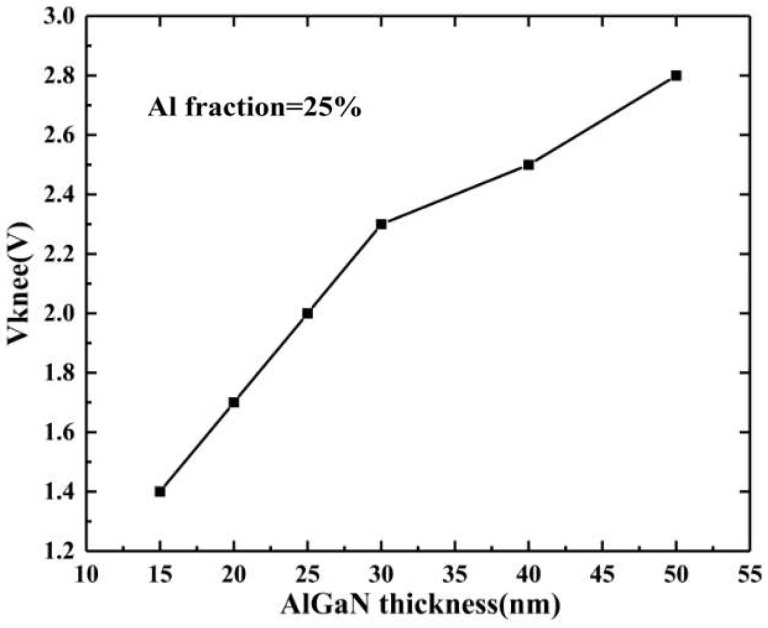
Variation of knee-point voltage with AlGaN thickness.

**Figure 5 micromachines-14-00168-f005:**
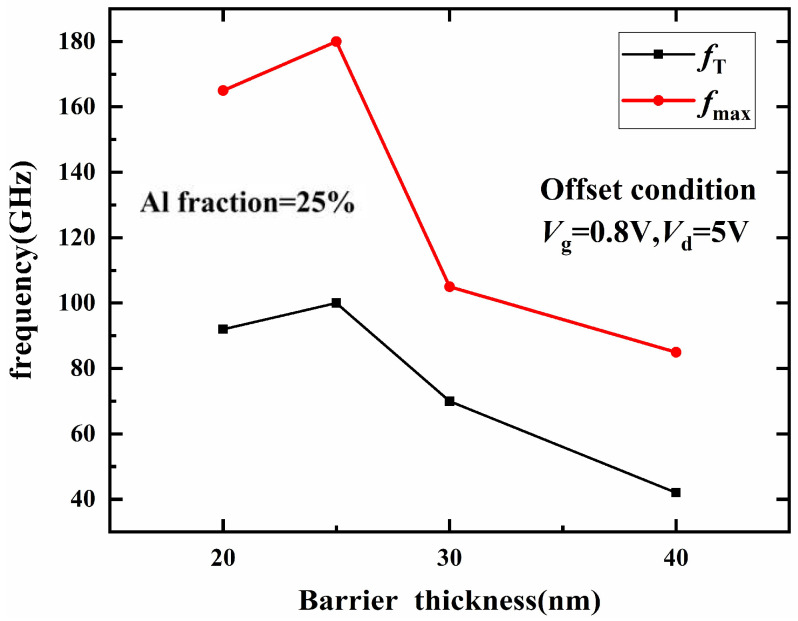
Effect of barrier thickness on *f_t_* and *f_max_*.

**Figure 6 micromachines-14-00168-f006:**
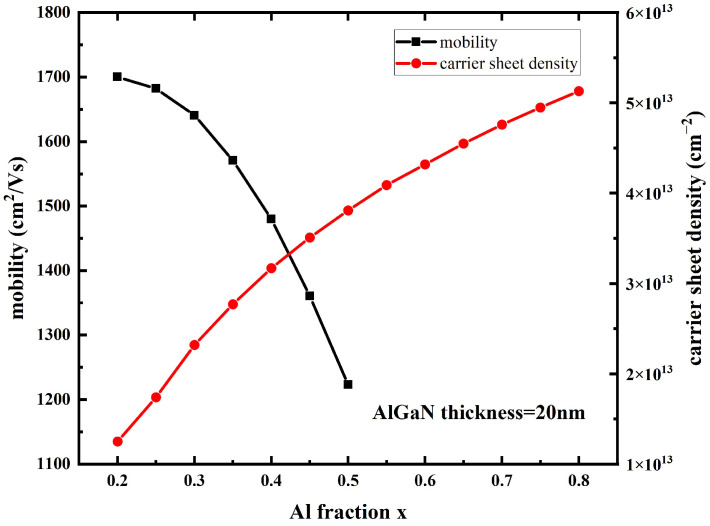
Mobility and carrier sheet density of devices with different Al components in barrier layers.

**Figure 7 micromachines-14-00168-f007:**
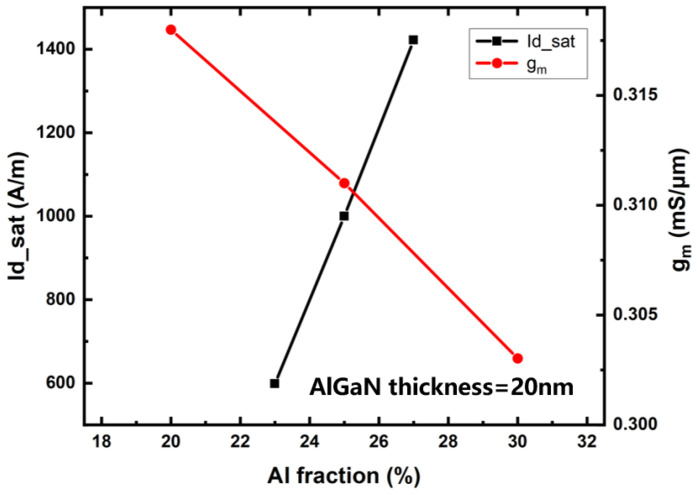
Simulation results of Al component on device saturation and transconductance.

**Figure 8 micromachines-14-00168-f008:**
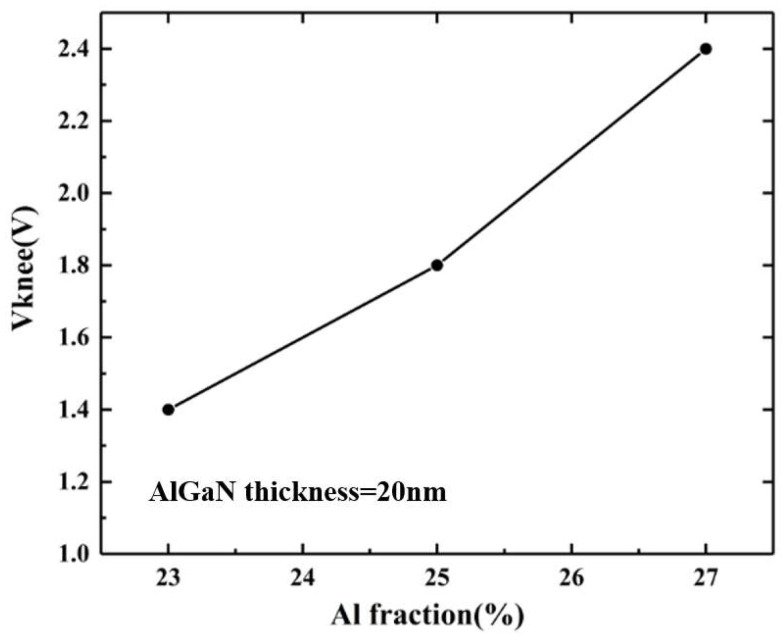
Variation curve of knee-point voltage of device with Al component of barrier layer.

**Figure 9 micromachines-14-00168-f009:**
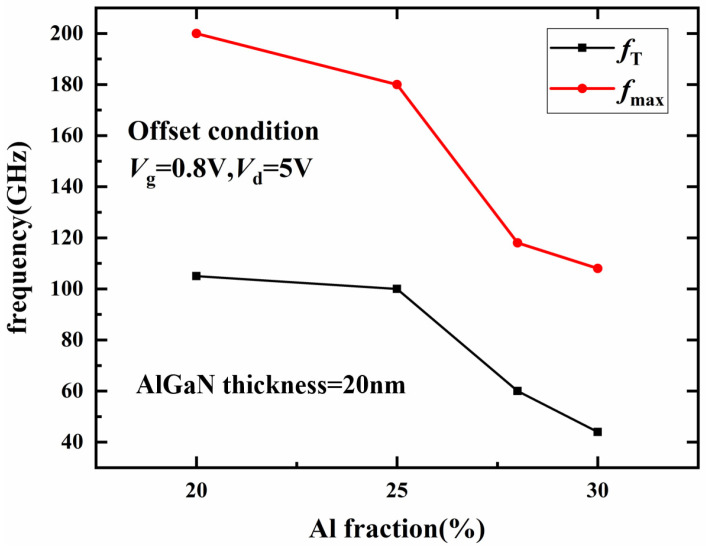
Influence of Al component of barrier layer on *f_t_* and *f_max_*.

**Figure 10 micromachines-14-00168-f010:**
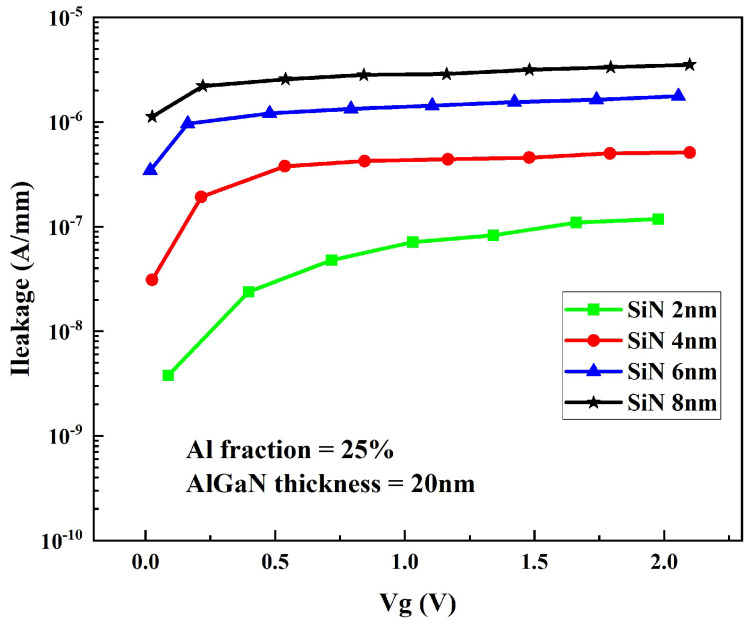
Variation curve of grid leakage current density with passivation layer thickness.

**Figure 11 micromachines-14-00168-f011:**
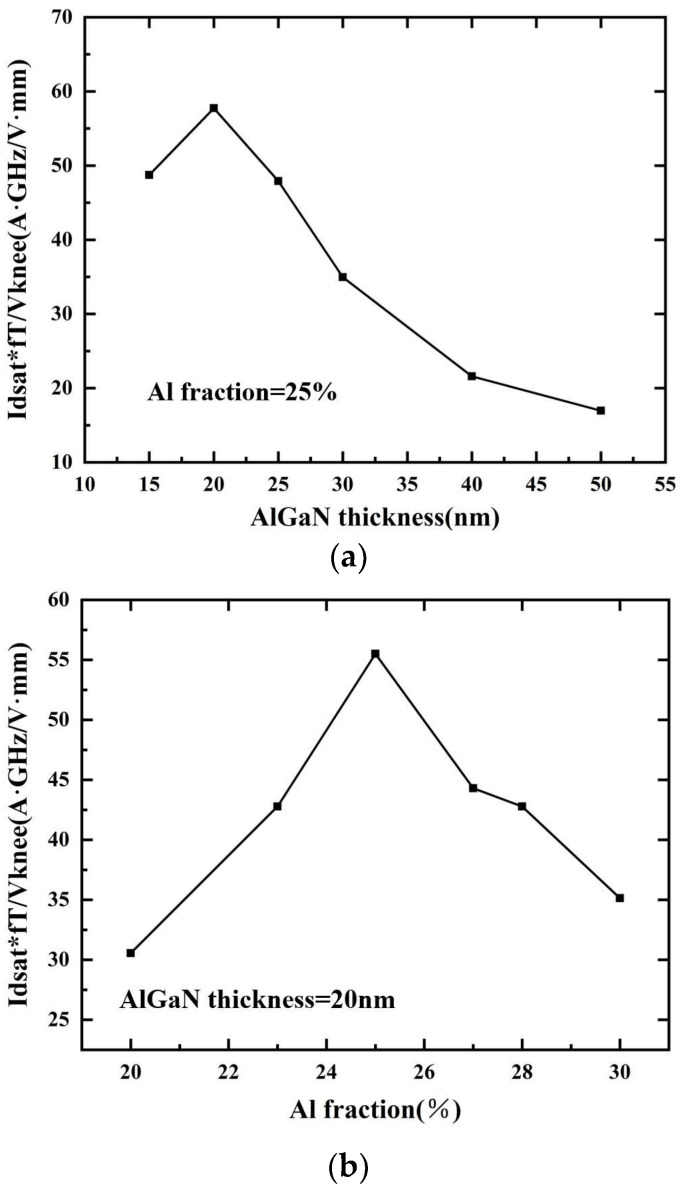
Calculation results of Optim values: (**a**) Optim values under different barrier layer thicknesses; (**b**) Optim values under different barrier layer Al fractions.

**Figure 12 micromachines-14-00168-f012:**
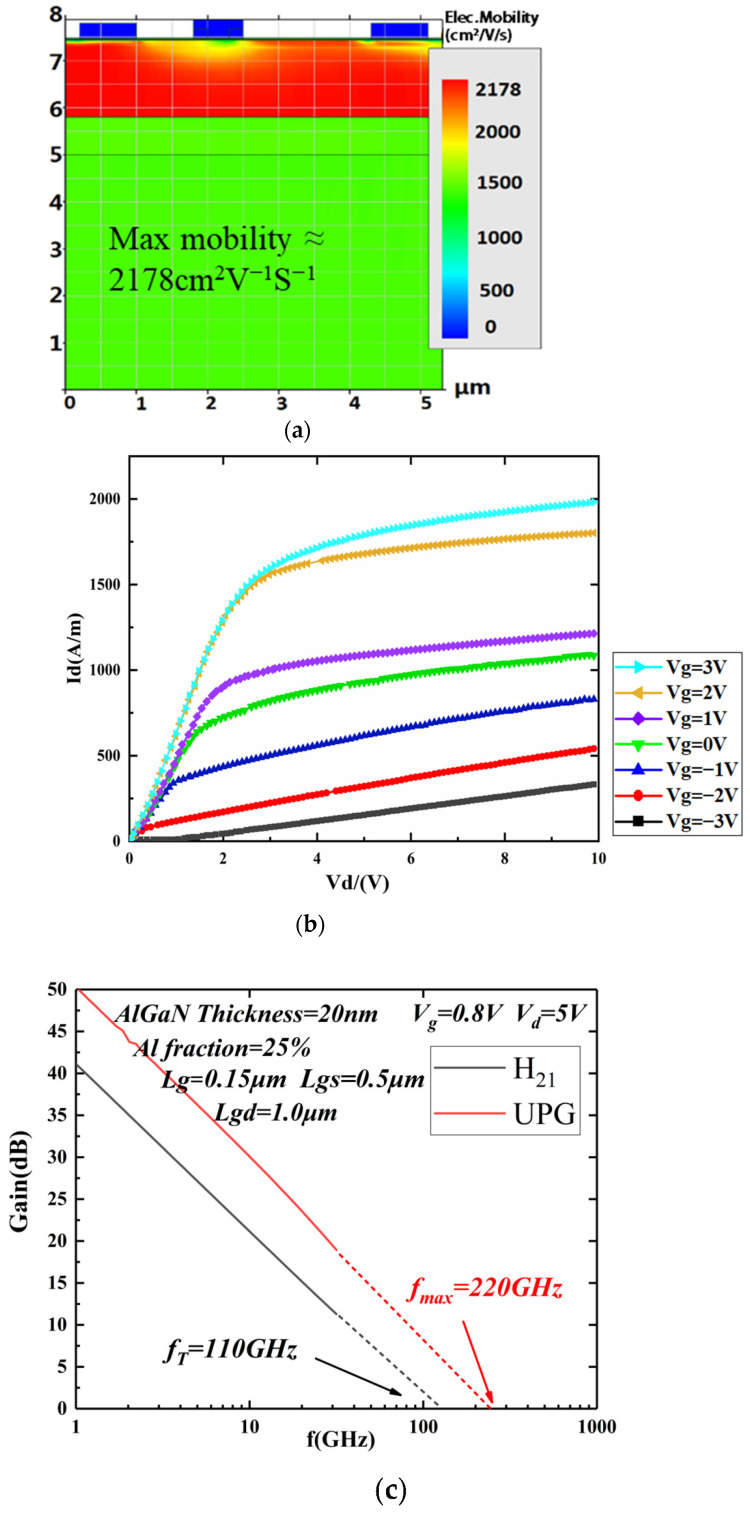
Device simulation results: (**a**) Mobility simulation and (**b**) output characteristics (**c**) *f_t_* and *f_max_*.

**Figure 13 micromachines-14-00168-f013:**
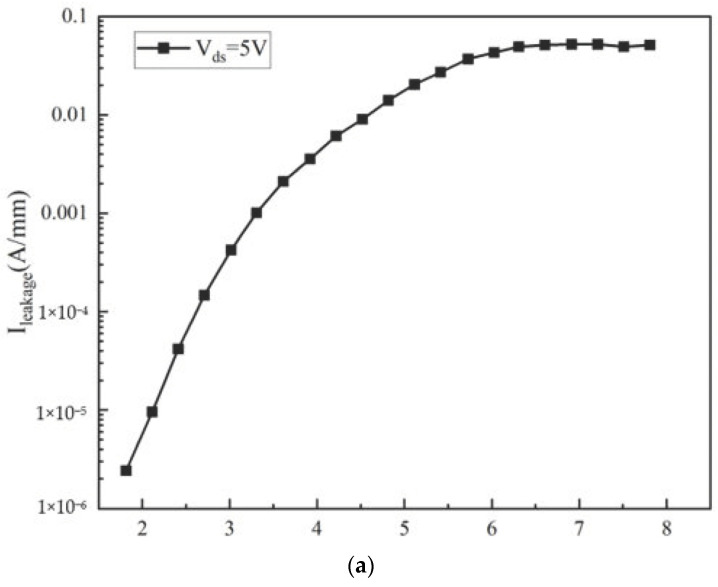
DC test result of the designed AlGaN/GaN HEMT device. (**a**) The gate leakage current test result, (**b**) the transmission curve test result.

**Table 1 micromachines-14-00168-t001:** HALL test results of the designed epitaxy.

Parameter	Mobility (cm^2^/V∙s)	Density (10^13^ cm^−2^)	Sheet Res (Ω/sq)
Test result	2063	1.86	161.41
Simulation result	2178	1.80	\

## Data Availability

Not applicable.

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
