# Peer review of "A Simulation Optimization Factor of Si(111)-Based AlGaN/GaN Epitaxy for High Frequency and Low-Voltage-Control High Electron Mobility Transistor Application"

_micromachines, 2023, doi:10.3390/mi14010168_

Round 1

Reviewer 1 Report

In this paper, authors have studied and simulated several essential physical parameters to determine the optimum operation of HEMT AlGaN/GaN epitaxy transistors on silicon substrates. The authors study the effect of AlGaN barrier thickness and Al concentration of barrier layer on the cut-off frequency ft, knee-point voltage Vknee and the saturation current Idsat. They defined an OPTIM = idsat*ft/Vknee parameter in order to optimize the barrier thickness and Al concentration.

Here are some remarks:

Line 58 : change 3:7 ratio by 0.3:0.7

Line 54 : change AlGan by AlGaN

Line 80 : Barriers thickness mentioned aren’t valid for all simulations. Change the sentence by putting from 10nm  to 50nm without specifying values.

 Figure 2 and 4 : change the name on abscissa “barrier thickness” by “AlGaN thickness”

 -Line 89 : the barrier thickness which decrease the electrical mobility start from 15 nm (see figure 2). I don’t understand why authors discuss of the decrease in electrical mobility from 30nm.

-Figure 3 : complete this figure by add some points simulation between 10 and 15 nm

-Figure 5 : complete the curve by add simulations points for 15nm, 22nm, 24nm, 26nm, 28nm. Barrier thickness between 20nm to 30nm is the most interesting and must be carefully analysed

-Figure 12b :  change the unit on abscissa unit Vd(V) not Vd(A/m)

-Figure 12c, in the frame, Al content indicated is 20%. Is it correct ? because in previous simulations Al concentration optimum is of 25%. You indicate Lg = 0.1um while previous results mention Lg=0.15um. What is the correct value used for simulation Lg=0.1um or 0.15um ?

 Regarding AlGaN barrier layers thickness simulations (figure 2, 3, 4, 5) no information is given on the Al concentration value. I assume that the Al concentration is 25%, is it correct ?

Same remark for Al concentration simulation (figure 7, 8, 9), It’s necessary to specify the barrier thickness value. I suppose in this case the value is 20nm.   

Reviewer 2 Report

The paper presents a simulation study on GaN HEMTs under different physical parameters, but the work lacks innovation and the simulation is not calibrated with experimental data, it is a pure simulation paper with limited reference value.

1.       The “Introduction” should be rewritten, as there is rarely any valuable information relevant to the study. It is all well known about the basics of GaN.

2.       The breakdown voltage or the device is not provided.

3.       The handling of the figures needs to be improved, e.g., the symbols in some figures do not conform to the specifications.

4.       There is no calibrated date and parameters are used in this paper. So, how to prove the validity of this simulation study?

5.       fT and fmax are obtained by what kind of simulation? The details need to be provided.

Round 2

Reviewer 2 Report

The revised version still has neither calibration data and parameters nor experimental results. It is still a pure simulation paper with limited reference value. It is recommended to wait for the experiments to be completed before submitting the manuscript.